# Effect of the Partial Substitution of Sodium Chloride on the Gel Properties and Flavor Quality of Unwashed Fish Mince Gels from Grass Carp

**DOI:** 10.3390/foods11040576

**Published:** 2022-02-17

**Authors:** Ruobing Pi, Gaojing Li, Shuai Zhuang, Qinye Yu, Yongkang Luo, Yuqing Tan, Ruitong Dai, Hui Hong

**Affiliations:** 1Beijing Laboratory for Food Quality and Safety, College of Food Science and Nutritional Engineering, China Agricultural University, Beijing 100083, China; piruobing1005@163.com (R.P.); ligaojing0724@outlook.com (G.L.); zhuangshuaicau@163.com (S.Z.); m13655709970@163.com (Q.Y.); luoyongkang@cau.edu.cn (Y.L.); yuqingtan@cau.edu.cn (Y.T.); dairuitong@gmail.com (R.D.); 2Center of Food Colloids and Delivery for Functionality, College of Food Science and Nutritional Engineering, China Agricultural University, Beijing 100083, China; 3Xinghua Industrial Research Centre for Food Science and Human Health, China Agricultural University, Xinghua 225700, China

**Keywords:** sodium salt substitutes, unwashed fish mince gels, gel characteristics, flavor quality

## Abstract

Excessive salt is usually required to maintain good gel properties and quality characteristics for unwashed fish mince gels (UFMG). This study aimed to investigate the effects of partial sodium chloride substitution (30%) with different substitutes (potassium chloride, disodium inosine-5′-monophosphate, basil) on the gel and flavor properties of UFMG from *Ctenopharyngodon idellus*. The results indicated that the texture and gel strength of NK (30% NaCl was replaced with 30% KCl) were fairly similar to that of N group (NaCl only), and the whiteness had improved significantly (*p* < 0.05), while the product eventually yielded a certain bitter taste. The addition of disodium inosine-5′-monophosphate (DIMP) significantly (*p* < 0.05) increased the hardness, chewiness, buriedness degree of tryptophan and gel strength, decreased the content of α-helix structure in the gels, while less change occurred in gel whiteness and network structure. Basil significantly (*p* < 0.05) reduced the buriedness degree of tryptophan, gel strength and whiteness, and deteriorated the gel structure. Nevertheless, the addition of DIMP or basil reduced the bitterness induced by KCl and improved the overall acceptability scores of gels of the N group. Moreover, there was no distinct difference in moisture content and water-holding capacity between all groups. Therefore, replacing sodium chloride in UFMG with 25% potassium chloride and 5% DIMP may be an ideal sodium salt substitution strategy.

## 1. Introduction

Fish mince is a restructured fish product that is processed from fillets or frames of raw fish meat without washing [1]. Such products better retain the original flavor and high-quality protein of fish, significantly lower water consumption and increase the product yield than washed surimi gels. Salt is most commonly used to solubilize and extract myofibrillar protein to obtain sticky exudates, then used to obtain gels [2]. The addition of salt not only adjusts the flavor by providing salty taste, but it also improves the texture, color and processing characteristics of surimi product. Compared with washed surimi gels, inferior frozen stability and gel strength was found in unwashed fish mince gels (UFMG) due to the higher content of active enzymes and their substrates, metals and lipids [3]. Therefore, UFMG always need more salt to reach an acceptable gel texture than surimi gels. However, excessive intake of sodium would increase the risk of hypertension and cardiovascular disease. Hence, effective strategies to reduce salt usage are essential to be promoted for fish products. Studies have shown that consumers are more inclined to accept the sodium salt substitutes with similar flavor and characteristics to sodium salt, or other food additives that can make up for the lack of saltiness [4].

Potassium chloride (KCl) is the most commonly used sodium salt substitute, which has properties very similar to sodium chloride (NaCl), in addition to anti-hypertensive properties and a higher recommended maximum intake than sodium [5]. KCl also has better muscle permeability than CaCl_2_ or MgCl_2_ [6]. When sodium is replaced by more than 35% KCl, the reduction of salt may lead to a negative impact on the sensory and physical and chemical properties of processed meat products [7,8]. Nevertheless, KCl imparts a bitter, metallic or chemical off-notes taste, which may result in an undesirable flavor to the surimi product. Hence, to elevate the sensory characteristics and overall acceptability of the food, it is always necessary to use KCl in combination with flavor enhancers or bitter blockers, including umami substances, such as sodium glutamate, 5′-adenosine monophosphate and 5′-inosine monophosphate [9], yeast extracts [10], hydrocolloids [11] or herbs and spices [4].

Inosine 5′-monophosphate (IMP) is the main umami compound in the meat of poultry and fish, which exerts a great influence on meat flavor formation [12]. The 5′-nucleotides and their salts, such as adenosine 5′-monophosphate (AMP), inosine 5′-monophosphate (IMP) and disodium inosine-5′-monophosphate (DIMP), have their inherent umami taste that is effectively used as bitter blocker, and they have been commonly used in food [13,14].

The use of herbal mixtures as sodium salt substitutes has also been confirmed to be a reliable way to reduce sodium intake, and the synergy between herbs always enhances the flavor of food simultaneously [4]. Seasonings derived from plants such as basil, oregano and rosemary could supplement the lack of salty taste with their inherent strong flavor. These seasonings can not only reduce the sodium salt content in food without compromising its microbial stability and flavor, but they also have a certain anti-hypertensive effect. However, uncertainty still exists about the application of sodium salt substitutes in the UFMG and the impact of a variety of sodium salt substitutes on physicochemical properties of UFMG.

Accordingly, the objectives of this study were to screen out a better sodium salt substitution method for UFMG. This study (1) investigated the effect of different sodium salt substitution methods on the sensory quality and gel properties of UFMG from grass carp; (2) analyzed the mechanism of the gel properties and the relationship between the gel characteristics with the internal structure of UFMG; (3) and evaluated the flavor characteristics and overall acceptability of UFMG with different sodium salt substitutes.

## 2. Materials and Methods

### 2.1. Materials

Ten fresh live grass carps with an average weight of 1500 ± 70 g were purchased from a seafood supermarket in Beijing and transported alive to the laboratory immediately. The sodium chloride, potassium chloride, disodium inosine-5′-monophosphate (DIMP) (Henan Jiaweiyuan Industrial Co., Ltd., Zhengzhou, China) and basil powder (Jiangsu Weitaimei Food Co., Ltd., Taizhou, China) used in the experiment were all food-grade additives.

### 2.2. Gel Sample Preparation

Alive grass carps were struck on the head to a fainted state and then gutted and filleted into fillets at once. After being washed with water (only to clean the bloodstains on the surface of the fish, not the water-soluble proteins), the white muscle was obtained by removing the skin, fin, scale, bone and dark muscle and stored at −20 °C for 18 h.

Many foods may be unpalatable when the substitution ratio of KCl exceeds 30%. According to the reported formulations of KCl and other substitutes [13,15], sodium salt substitutes with different formulations were set with some adjustment, as shown in Table 1. The additive usage and dosage accorded with the edible standard and have been successfully used in a variety of foods [2,3,13], which could be applied to actual production and consumption. Sodium salt substitutes were mixed with grass carp fish mince, water (the ratio of meat to water of 3:2) and 4% starch. The mince of the same weight was centrifuged under 1255 g by a TGL-20M medical centrifuge (Changsha Pingfan Instruments Co., Ltd., Changsha, China) in order to ensure the uniform texture of the subsequently prepared gels and to prevent the presence of holes in the gels, thereby affecting the evaluation of gel quality. After that, the gels were obtained after a two-step heating process, firstly heated at 45 °C for 20 min and then at 90 °C for 15 min in HH-W600 thermostat water bath (Jintan Science Analysis Instrument Co., Ltd., Jintan, China). Afterward, the gels were chilled in ice water immediately for further analysis.

### 2.3. Gel Properties

#### 2.3.1. Textural Properties

UFMG samples were cut into equal-height cylinders with a diameter of 15 mm. The textural properties of gels, including hardness, springiness, cohesiveness, chewiness and adhesiveness, were assessed with a CT-3 texture analyzer (Brookfield, WI, USA) according to the method of [16] with some modification. At the textural profile analysis (TPA) mode, each sample was tested at the 40% deformation rate and moved at the test speed of 1 mm/s using a TA3/100 square probe.

#### 2.3.2. Gel Strength

Breaking force (g) and breaking distance (mm) of the gels were measured by a CT-3 texture analyzer equipped with a TA50 spherical probe [17]. Gel properties were obtained from a single compression mode at 50% deformation, 7.0 g trigger force and 0.5 mm/s test speed using a spherical head plunger (5 mm). Gel strength was obtained by the multiplication of breaking force (g) and breaking distance (mm).

### 2.4. Whiteness

The samples were cut into 10 mm thick slices, and L* (lightness), a* (redness) and b* (yellowness) of gels were measured using an NS800 spectrophotometer (3 nh, Shenzhen, China). Then, whiteness was calculated according to following formula [18]:$$\mathrm{Whiteness}=100-\sqrt{{\left(100-L^{*}\right)}^{2}+a^{{*}^{2}}+b^{{*}^{2}}}$$

### 2.5. Water-Holding Capacity and Moisture Content

Water-holding capacity (WHC) was determined as previously reported, with slight modifications [19]. The prepared UFMG were cut into cylinders of the same size and weighed in the centrifuge tube (M1). Then, the sample was wrapped with three layers of filter paper and weighed again after centrifugation at 7000× *g* (M2). The results were expressed as the weight percentage of the liquid retained in the sample and calculated according to the equation below:$$\mathrm{WHC}\;(\%)=100-\left[\left\{\left(M_{1}-M_{2}\right){/M}_{1}\right\}\times 100\right]$$

The moisture content in UFMG was determined by an XY-100MS moisture analyzer (Xingyun Electronic Equipment Co., Ltd., Changzhou, China). Gels with the same weight were placed in the sample pan, respectively, then closed in the heating chamber for automatic drying. The moisture content values were defined as the ratio of weight difference before and after drying to the original weight.

### 2.6. Raman Spectrometric Measurements

Raman spectroscopy analysis of the samples was determined as the method described by [20] with some modifications. The samples were scanned using the Raman spectrometer (LabRAM HR Evolution, HORIBA, Ltd., Kyoto, Japan) equipped with a 532 nm laser as the excitation source. The Raman spectrometer was run at 100% power and 60 s acquisition time in the range of 400–3100 cm^−1^. The deconvolution of the curve was completed using Peakfit V4.12 software (Systat Software Inc., San Jose, CA, USA).

### 2.7. Gel Microstructure

UFMG were fixed for 24 h at 4 °C in 2.5% glutaraldehyde solution first, then dehydrated in incremental concentrations of ethanol [21]. The dried specimens were sputter-coated with gold and examined under a JSM-7001F field emission scanning electron microscope (JEOL Ltd., Tokyo, Japan) at an acceleration voltage of 30 kV.

### 2.8. Electronic Tongue Analysis

Electric tongue analysis was performed following [22], with some modifications. About 15.0 g UFMG were homogenized with 100 mL deionized water; the supernatant was extracted after centrifugation 6558× *g* at 4 °C, and determined by an electronic tongue system (SA402B; Tokyo, Japan) equipped with five chemical sensors for sourness (CA0), bitterness (C00), saltiness (CT0), umami (AAE) and astringency (AE1), and two reference electrodes. All electrodes were immersed in the corresponding soaking liquid for 24 h before the measurements.

### 2.9. Sensory Evaluation

Sensory evaluation was performed using acceptability test to evaluate the acceptability of the UFMG. Sixty students evaluated four UFMG from eight dimensions, namely seven sensory attributes (color, organization structure, odor, saltiness, bitterness, springiness, chewiness and taste) [23] and a comprehensive score based on all attributes.

UFMG were provided to each member at the same time after reaching room temperature. After tasting each sample, the team members rated each group of samples and scored them.

### 2.10. Statistical Analysis

Experimental data from replications were analyzed by one-way ANOVA analysis using SPSS 20 software (SPSS Inc., Chicago, IL, USA), and all graphs were mapped with Origin 2018. The differences between measurements were performed by Duncan ‘s test within a 95% confidence interval.

## 3. Results and Discussion

### 3.1. Gel Properties Analysis

#### 3.1.1. Gel Textural Properties

In the process of thermal gelation, myosin undergoes irreversible structural changes, finally leading to a stable three-dimensional network structure. The level of salt influences the amount of exuded protein obtained and plays a role in binding, affecting the mechanism of protein aggregation [2,24]. As shown in Figure 1a,b, there was no remarkable difference between the texture characteristics in N and NK, which indicated that KCl induced little effect on the aggregation of myofibrillar protein. Additionally, KCl could achieve 30% sodium salt substitution without altering the texture characteristics of UFMG significantly. Similar results were also confirmed in reduced-fat mortadella [25] and fermented cooked sausages [26]. The gel hardness and gumminess of NKI increased significantly (*p* < 0.05) to 384.2 g and 250.60 g compared to other treatments. Likewise, the hardness of low-sodium fermented sausages was significantly (*p* < 0.05) increased after the addition of IMP compared with the sausages with 50% potassium chloride substitution [14]. The results demonstrated that DIMP was efficient in improving the hardness of UFMG. This phenomenon might be related to the partially similar structure of DIMP with phosphates. Sodium pyrophosphate and sodium tripolyphosphate could significantly enhance the hardness and springiness of the gel [27]. Therefore, it was speculated that DIMP also had an impact on improving the texture properties of UFMG to some extent. However, due to longer chain length, this enhancement was not as intense as other phosphates [28]. Compared with the N group, the addition of basil undermined the chewiness and gumminess of UFMG significantly (*p* < 0.05), suggesting a decrease in salt content, and the fiber of basil triggered inadequate unfolding of myosin molecules, thereby affecting the formation of the continuous gel network and altering the final texture of the gel [29].

#### 3.1.2. Gel Strength

Less salt has always resulted in less content of dissolved myofibril protein and had a negative impact on the gelation of proteins, thus lowering gel strength of the restructured fish products, such as surimi [5]. The UFMG in N and NK had the most similar gel strength in Figure 1c. The results may be explained by the lower substitution ratio, and sodium and potassium are elements of the same main group; therefore, their chloride salts have ionic strength, similar molecular structures and functional properties. Meanwhile, the UFMG in NKI exhibited the strongest gel strength due to being positively correlated with the increase in gel hardness, which agreed with the result in Figure 1a. Nevertheless, gel strength significantly (*p* < 0.05) decreased to 299.54 g·mm due to the addition of basil. Basil particles did not induce distinctive changes in gel hardness but still obstructed the crosslink of myofibrillar proteins, consequently leading to a negative impact on the mechanical properties of UFMG [5,30].

### 3.2. Whiteness, WHC and Moisture Content

Changes in the color of unwashed grass carp gels (L*, a* and b*) depend on different methods of sodium salt substitution (Table 2). All of the a* were negative in UFMG, which indicated that all samples were greener than the standard plate. Based on the results, the L*, a* and b* of NK were significantly (*p* < 0.05) higher than the N group, while the yellowness value increased significantly (*p* < 0.05). The work of Zhao et al. (2021) demonstrated that the L*, a* and b* of sodium-reduced chicken sausage appeared to have analogical variation tendency, but there was no marked difference in the other values, except the value in the early storage period [31]. Greiff et al. (2015) found an increase in whiteness and a decrease in color hue with decreased salt content [32]. Therefore, it was speculated that the results may be caused by varied substitution ratio and uneven fish color.

Additionally, the L* and whiteness values of the NKB group apparently dropped to 73.80 and 80.28, respectively, with a huge rise in b*, which exhibited huge alteration with those gels in the N group. The consequence illuminated that the color of basil itself would induce a sharp increase in b*, eventually leading to a striking fall in the whiteness of UFMG. The large sensory color difference caused by basil may decrease the sensory acceptance of the UFMG in NKB group. Interestingly, the L*, a*, b* and whiteness values of the gels in the NKI group had indistinctive variance with the N group, indicating that the sensory color attributes of these two groups are relatively close. Thus, the UFMG of NKI group may be more likely to be accepted by consumers.

Stable and orderly network structure plays an important role in preventing moisture and fat loss of gel, lower breaking force usually being cohesive with lower water holding capacity (WHC) [33]. The water content of UFMG showed an unremarkable difference between the four treatments. Additionally, the WHC of NKB slightly dropped to 81.27%, which was consistent with the lower breaking force and deformation results of NKB. Analogical conclusion was manifested in the salted pork when the substitution rate of KCl was less than 70% [34]. Furthermore, adding KCl with the same molar concentration as NaCl had minor or almost no effect on moisture, cooking loss and WHC in cooked meat [35]. Therefore, it could be inferred that the substitution of sodium salt with basil induced a slight decline in the WHC. However, lower sodium substitution rate (30%) had no excessive effect on the formation of gel network with great WHC.

### 3.3. Raman Spectra Analysis

Figure 2a showed the Raman spectrum of UFMG at 400–3100 cm^−1^. During the formation of UFMG, protein molecules aggregate with the increasing temperature, the α-helix structure unfolds and gradually transforms into β-sheets, β-turns and random coils [19,36]. The secondary structure information of the protein was revealed by the amide I band (1600–1700 cm^−1^) in the Raman spectrum. Figure 2b demonstrated the results obtained by the amide I band after deconvolving and calculating the relative content of each secondary structure. The discrepancy of secondary structure ratios would lead to different peak shapes of the amide I band, α-helix corresponding to the band located around 1650–1660 cm^−1^ [37,38]. From the results shown in Figure 3, compared with the N group, the α-helix content of UFMG was significantly decreased from 25.73% to 22.04% (*p* < 0.05) when DIMP was added to UFMG. On the contrary, the addition of basil resulted in a significant increase in the content of α-helix to 28.02% (*p* < 0.05). The secondary structure of protein is well correlated with the gel strength; higher content of the α-helix structure usually has a negative impact on the gel strength, while more β-sheets, β-turns and random coils induce a reinforcement in the gel strength [39,40]. Therefore, it can be inferred that the addition of DIMP promoted the transition from α-helix to β-turn, thus forming a network with higher gel strength. However, the addition of basil led to a destruction of gel network structure, which was consistent with the results obtained in Figure 2.

This might be caused by the discrepancy between different secondary structures. Compared with the compact structure of β-sheet, the α-helix structure has a relatively small surface area and strong hydration strength [36]. β-turns are usually composed of four amino acid residues with a hydrogen bond formed between the first and fourth amino acid residues. The decreased β-turn content indicated that the hydrogen-bonding network was broken [41], and hydrogen bond is the primary chemical bond to maintain the secondary structure of the protein; therefore, it could be speculated that the gel strength was disparate, according to different relative contents of secondary structure in the protein [42].

### 3.4. Tryptophan Microenvironment

An important feature of tryptophan residues was the doublet Raman bands at 1360 cm^−1^ and 1340 cm^−1^, the intensity ratio of the doublet (I1360/I1340) reflecting the transition of tryptophan microenvironment. A higher ratio indicates an increased buriedness of tryptophan, while a lower ratio exhibits that they are exposed to a hydrophilic environment [43]. Compared with the N group, the doublet ratio of NKI group exhibited a significant rise from 0.58 to 0.66 (*p* < 0.05), which suggested that the buriedness degree of tryptophan increased when DIMP was added into the UFMG (Figure 4). The results illustrated that DIMP better promoted protein aggregation, more likely via hydrophobic interaction, and contributed to the formation of a more ordered gel structure than other groups. On the contrary, the ratio significantly decreased to 0.50 after adding basil to the UFMG (*p* < 0.05), which meant that more tryptophan residues were exposed. This might because of the partial loss of water, which induced a destruction of the hydrophobic cluster structure and weakened the gel network to a certain extent [44]. Therefore, the transform in the tryptophan microenvironment and protein secondary structure led to changes in gel properties [45].

### 3.5. Gel Microstructure

In Figure 5, the UFMG microstructures in NK and NKI were relatively dense without obvious pores, and the substitutes had no apparent effect on the formation of gel structure compared with the N group. Same as sodium chloride, potassium chloride not only had the same ionic strength, but it also had a certain ability to promote the dissolution of myofibril protein, subsequently forming a relatively stable gel structure [46]. Meanwhile, the low substitution ratio of DIMP and the existence of small amount of sodium ions may account for the indistinctive gel structure change. Hence, NK and NKI were able to achieve a good substitution effect without affecting the UFMG structure under 30% substitution ratio. However, the UFMG microstructure in the NKB group showed more and deeper pores and then collapsed, which was consistent with the decrease in gel strength of the NKB in Figure 2. The reason might be that basil granules restrained the arrangement and aggregation behavior of myofibrillar proteins, thereby destroying the gel structure, directly leading to a decrease in the gel strength of NKB.

### 3.6. Gel Flavor

#### 3.6.1. Electronic Tongue Analysis

Electronic tongue was designed to objectively evaluate the overall information of taste in the sample with a series of non-specific electrochemical sensors [47]. Figure 6 indicates the discrepancy in sourness, bitterness, astringency, aftertaste-B, aftertaste-A, umami, richness and saltiness of UFMG under different sodium salt substitution ways. As shown in Figure 6, the flavors of different groups were relatively similar, except for the bitter taste. Compared with the N group, the bitterness of NK was significantly (*p* < 0.05) increased, and the UFMG bitterness was significantly (*p* < 0.05) reduced after the addition of DIMP or basil. Nevertheless, there were similar salty tastes among different samples, which may be explained by the low substitution ratio of DIMP and basil, and more importantly, the saltiness is mainly provided by chloride ions.

Furthermore, what stood out in the table was that the addition of potassium chloride conspicuously increased the bitterness of samples. However, the bitterness was significantly (*p* < 0.05) reduced from 10.15 to 9.64 and further down to 9.59 after the addition of DIMP and basil, respectively. DIMP imparted an umami taste by the phosphate ester on the 5′carbon of the ribose moiety and purine moiety with a hydroxyl function on the 6′carbon of the purine ring [48]. Additionally, umami substances such as DIMP can work in tandem with salt taste and inhibit the perception of sourness or bitterness with less total sodium and fewer calories [49]. Similarly, basil has a special volatile flavor, so it can partially cover up bitterness and enhance the characteristic flavor of UFMG. Overall, the NKI and NKB groups had acquired relatively greater substitution effects in terms of flavor.

#### 3.6.2. Sensory Evaluation

In terms of color, no remarkable difference was found between the other three groups, except that the score of the NKB group was significantly dropped (*p* < 0.05) (Figure 7). The results indicated that most sensory evaluation panelists had a clear perception of the color alteration caused by basil rather than potassium chloride. Additionally, the NKI group and the NKB group had the best and worst organizational structures among the samples, respectively, which is consistent with the results obtained from Figure 5. As for flavor scores, the taste and aroma of basil are mainly caused by monoterpenes and phenylpropanoids [50], including linalool, eugenol, chavicol and other substances [51]. These compositions render a relatively strong aromatic odor, and thus rendered UFMG a unique scent and flavor characteristics. Thereby, the highest odor score and taste score were obtained among these four groups after the addition of basil. However, the addition of potassium chloride led to lower taste scores (5.37) in the NK group. A possible explanation for this might be that the bitter and metallic taste induced by KCl caused a slight decrease in the salty sense experienced by the testers, which was consistent with the higher bitterness intensity obtained in Figure 6 and the slight increase in the bitterness score of the NK group. Overall taste is affected by the comprehensive effect of multiple flavors in the UFMG instead of one sort of flavor. Combined with the results of electronic tongue analysis and sensory evaluation, although the addition of KCl caused a certain bitter taste, under the influence of other tastes, the role of bitterness in affecting the panelists’ perception of the overall taste may have been weakened, resulting in a slight bitterness that may not have been perceived by some panelists. After considering all the factors comprehensively, the most striking result to emerge from the data was that NKI and NKB gained higher overall acceptability scores of 7.18 and 7.19 than the N group. Therefore, it can be speculated that these two methods may potentially provide effective sodium salt substitution means in UFMG.

On the whole, most sensory evaluation scores of N and NK were similar, except for a slight reduction in bitterness and saltiness scores. However, when compared with the N group, the NKI group had better tissue structure, hardness, elasticity and chewiness scores, as well as superior taste and overall acceptability scores, in addition to the salty sense that was not inferior to the N group.

Hence, it could conceivably be hypothesized that it may be a feasible way to partially substitute sodium salt in UFMG, which would have certain reference implications for the processing and production of UFMG. Moreover, the NKB samples had the most excellent taste score and overall acceptability among four groups, which indicated that the vast majority of testers preferred the flavor changes caused by basil without undesirable feelings, but it cannot be ruled out that changes in the appearance of UFMG may have affected panelists’ feelings. In addition, panelists’ acceptance of the basil flavor is also related to some factors, such as panelists’ personal preferences and the frequency of exposure to basil.

## 4. Conclusions

The results suggested that the unwashed fish mince gels added with DIMP had better texture characteristics and gel properties without significant changes in gel microstructure and WHC. In terms of flavor, Basil and DIMP were both able to mitigate the flavor deterioration caused by KCl and were even able to achieve a higher sensory score than the N group. Therefore, replacing 30% NaCl with 25% KCl and 5% DIMP is a feasible way to reduce sodium content in unwashed grass carp mince gels. Additionally, it may also have potential commercial application value in other fish or meat mince gels.

## Figures and Tables

**Figure 1 foods-11-00576-f001:**
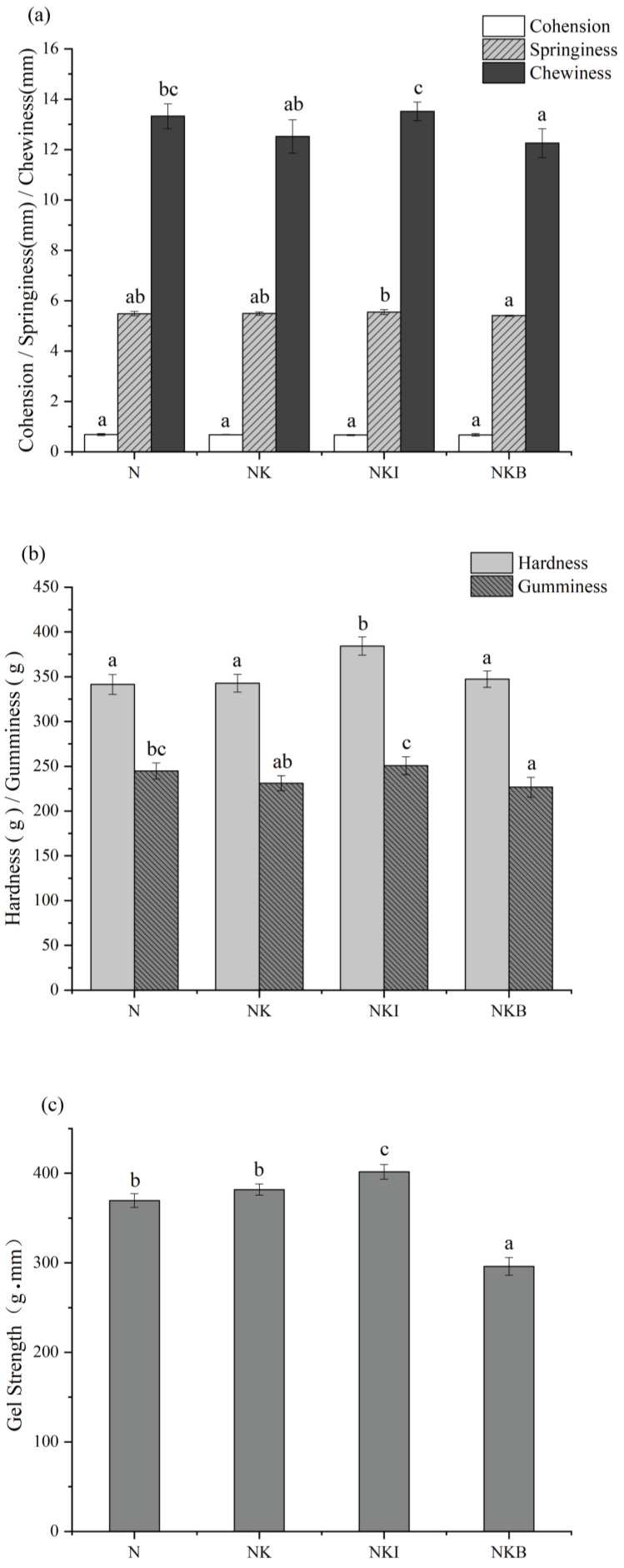
Gel properties of (**a**) cohesiveness, springiness, chewiness (**b**) hardness, gumminess and (**c**) gel strength of N (NaCl only), NK (30% NaCl was replaced with 30% KCl), NKI (30% NaCl was replaced with 25% KCl and 5% DIMP), NKB (30% NaCl was replaced with 25% KCl and 5% basil). DIMP stands for disodium inosine-5′-monophosphate. Different letters (a–c) indicate significant differences (*p* < 0.05) between the samples.

**Figure 2 foods-11-00576-f002:**
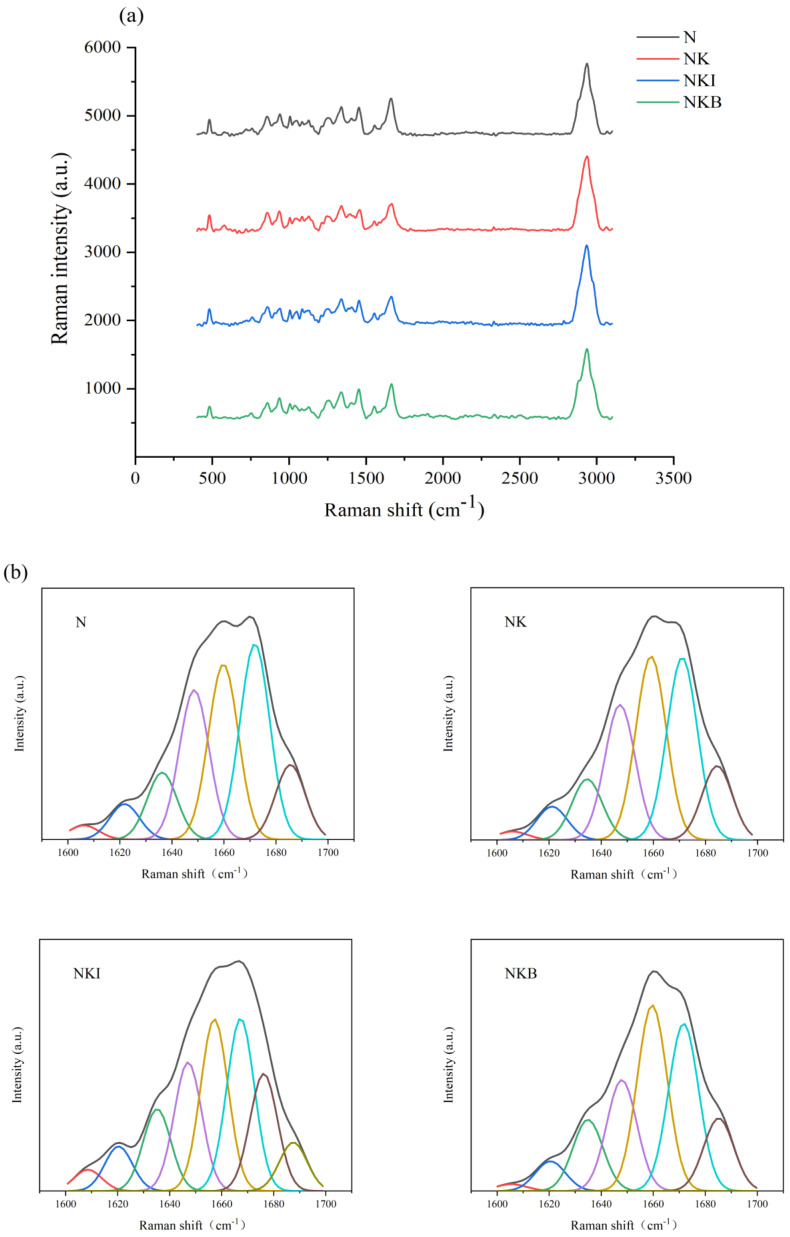
Raman spectrum (**a**) and deconvoluted amide I Raman band (**b**) of N (NaCl only), NK (30% NaCl was replaced with 30% KCl), NKI (30% NaCl was replaced with 25% KCl and 5% DIMP), NKB (30% NaCl was replaced with 25% KCl and 5% basil). DIMP: disodium inosine-5′-monophosphate.

**Figure 3 foods-11-00576-f003:**
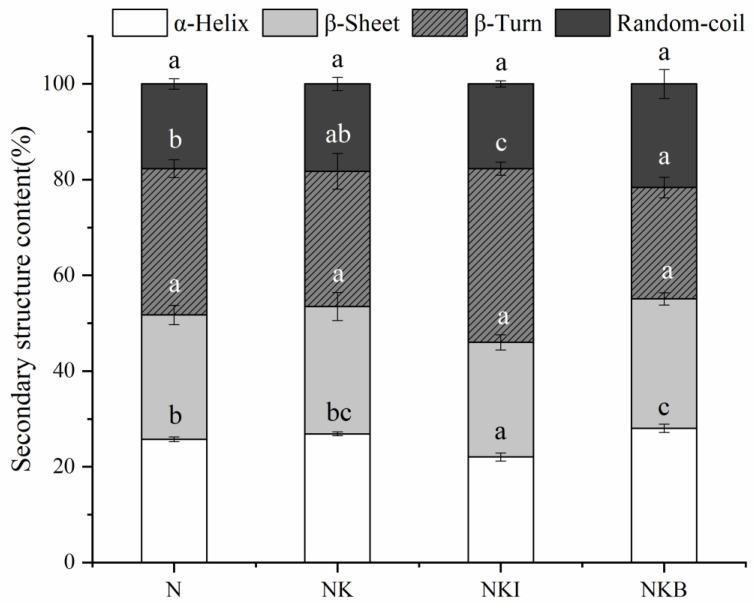
Secondary structure content of N (NaCl only), NK (30% NaCl was replaced with 30% KCl), NKI (30% NaCl was replaced with 25% KCl and 5% DIMP), NKB (30% NaCl was replaced with 25% KCl and 5% basil). DIMP stands fordisodium inosine-5′-monophosphate. Different letters (a–c) indicate significant differences (*p* < 0.05) between the samples.

**Figure 4 foods-11-00576-f004:**
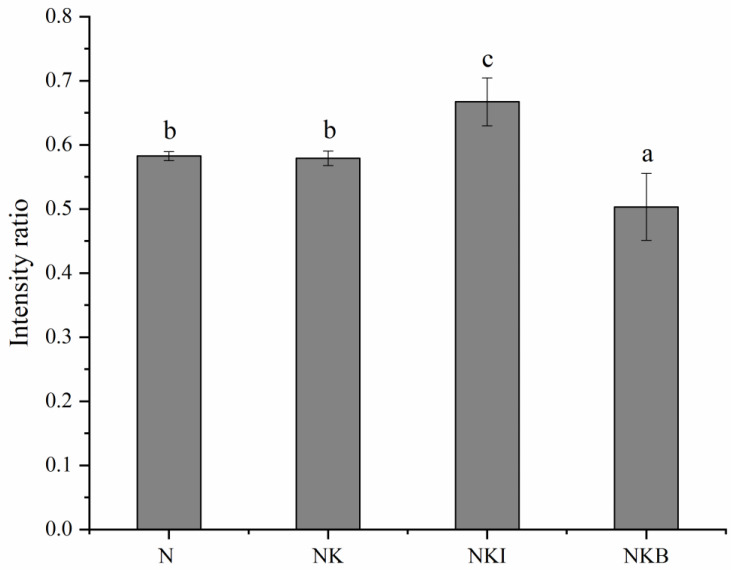
Tryptophan doublet ratio of N (NaCl only), NK (30% NaCl was replaced with 30% KCl), NKI (30% NaCl was replaced with 25% KCl and 5% DIMP), NKB (30% NaCl was replaced with 25% KCl and 5% basil). DIMP stands for disodium inosine-5′-monophosphate. Different letters (a–c) indicate significant differences (*p* < 0.05) between the samples.

**Figure 5 foods-11-00576-f005:**
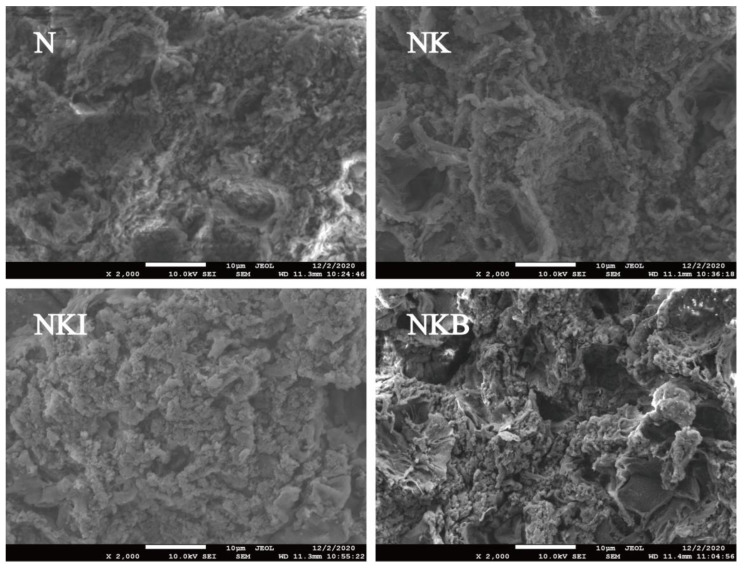
Gel microstructure (×2000) of N (NaCl only), NK (30% NaCl was replaced with 30% KCl), NKI (30% NaCl was replaced with 25% KCl and 5% DIMP), NKB (30% NaCl was replaced with 25% KCl and 5% basil). DIMP stands fordisodium inosine-5′-monophosphate. Different letters (a–c) indicate significant differences (*p* < 0.05) between the samples.

**Figure 6 foods-11-00576-f006:**
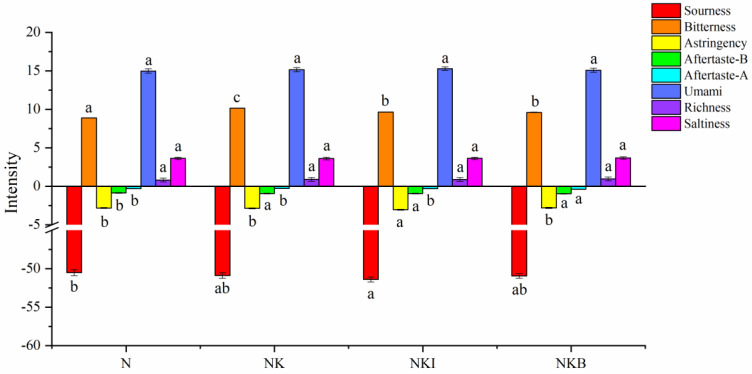
Bar plot for electronic tongue data of N (NaCl only), NK (30% NaCl was replaced with 30% KCl), NKI (30% NaCl was replaced with 25% KCl and 5% DIMP), NKB (30% NaCl was replaced with 25% KCl and 5% basil). DIMP stands for disodium inosine-5′-monophosphate. Different letters (a–c) indicate significant differences (*p* < 0.05) between the samples.

**Figure 7 foods-11-00576-f007:**
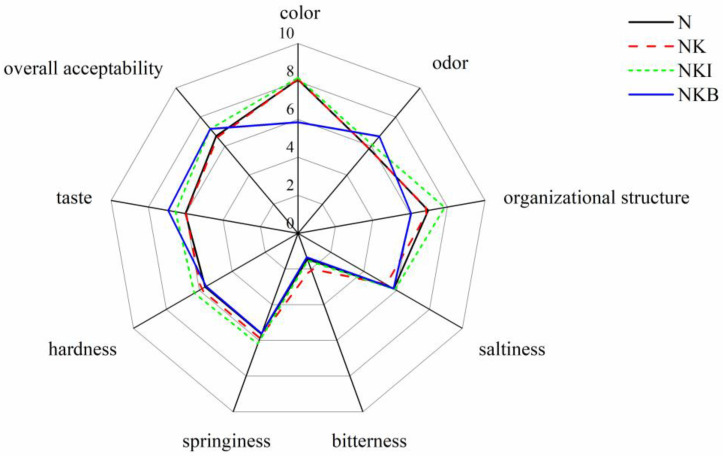
Sensory scores of N (NaCl only), NK (30% NaCl was replaced with 30% KCl), NKI (30% NaCl was replaced with 25% KCl and 5% DIMP), NKB (30% NaCl was replaced with 25% KCl and 5% basil). DIMP stands for disodium inosine-5′-monophosphate.

**Table 1 foods-11-00576-t001:** Compositions of sodium salt substitutes for preparation of UFMG.

Group	NaCl	KCl	DIMP	Basil
N	1.80%	-	-	-
NK	1.26%	0.54%	-	-
NKI	1.26%	0.45%	0.09%	-
NKB	1.26%	0.45%	-	0.09%

N (NaCl only), NK (30% NaCl was replaced with 30% KCl), NKI (30% NaCl was replaced with 25% KCl and 5% DIMP), NKB (30% NaCl was replaced with 25% KCl and 5% basil), DIMP: disodium inosine-5′-monophosphate. “-” indicates that this substance was not added to this group of substitutes.

**Table 2 foods-11-00576-t002:** Whiteness, WHC and moisture content of fish mince gels.

Group	L*	a*	b*	Whiteness	WHC (%)	Moisture Content (%)
N	82.62 ± 0.22b	−2.77 ± 0.02a	4.33 ± 0.07a	81.88 ± 0.22b	83.12 ± 0.54a	81.16 ± 0.42a
NK	83.64 ± 0.13c	−2.53 ± 0.02b	4.77 ± 0.04b	82.77 ± 0.12c	83.17 ± 0.69a	81.09 ± 1.65a
NKI	82.44 ± 0.17b	−2.81 ± 0.02a	4.54 ± 0.15a	81.65 ± 0.13b	83.58 ± 1.33a	81.41 ± 0.42a
NKB	81.18 ± 0.10a	−2.18 ± 0.03c	5.59 ± 0.09c	80.25 ± 0.12a	81.27 ± 1.52a	81.27 ± 1.64a

N (control group), NK (30% NaCl was replaced with 30% KCl), NKI (30% NaCl was replaced with 25% KCl and 5% DIMP), NKB (30% NaCl was replaced with 25% KCl and 5% basil). DIMP stands for disodium inosine-5′-monophosphate. WHC indicates water holding capacity. Different letters (a–c) indicate significant differences (*p* < 0.05) between the samples.

## Data Availability

Data are contained within the article.

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
