# Peer review of "Effect of the Partial Substitution of Sodium Chloride on the Gel Properties and Flavor Quality of Unwashed Fish Mince Gels from Grass Carp"

_foods, 2022, doi:10.3390/foods11040576_

Round 1

Reviewer 1 Report

The manuscript presents novelty information about the effect of partial substitution of NaCl to produce low salt gel from carp.

I suggest some minor revisions as follow:

Introduction:

Line 49 change to used

Lines 73 to 79: please rewrite the aims. The three aims proposed here look like activities more than objectives. The title is closer to the aim expected for this work

Line 75: what do you refer to as “alternative methods”?

Methodology

Line 95: Was the gel obtained after centrifugation?, please clarify this point

Results and discussion

Lines 198-200: please rewrite this paragraph because it’s hard to follow

Line 306: delete “obvious”

Line 318: …UFMG under different alternative methods… What do you refer to with the “alternative method”?

Line 382: Why was replacing 30%NaCL with 25%KCl and 5% DIMP considered “might be feasible” and not as feasible?

Reviewer 2 Report

The work foods-1516211 studied the effects of partial NaCl substitution with different substitutes on the gel and flavor properties of unwashed fish mince gels (UFMG).

The manuscript is well written and describes a considerable amount of evidence that provides scientific support. However, I believe that more specifications are missing in the M&M section. In addition, I think the authors should clarify the commercial viability of the formulations studied, otherwise, the study would have no applicability.

Below, I leave some issues that must be clarified for the manuscript.

- L49: “used” should be corrected.

- The authors do not provide details of the equipment used in M&M, e.g., centrifuge, heaters, chillers, etc. I recommend indicating at least the brand and reference.

- The abbreviations used in Table 1 must be specified before, or they should be indicated as a footer.

- An explanation should be given for the formulations listed in Table 1.

- The authors spend much of the R&D on giving details of statistical significance, without discussing its practical meaning. The interpretation of the p values is well known and there is no need to spend so much manuscript space repeating these results already made clear in Tables and Figures. This is a mistake that is often made and which I recommend authors avoid since this section should be devoted to interpreting these significances in practical terms, without the need to reiterate them in the text.

- The authors must also provide legal support to show that the formulations used in Table 1 can be used for human consumption.

-183-186: Adhesiveness is not explained in Figure 1 or the manuscript.

- It should be clarified when and why we talk about "improvements" when explaining the results in Figure 1.

- Support should be given for the results and theories expressed by the authors, e.g., L188-192.

- Only significant figures should be expressed in Table 2.

-L292: Support with previous work.

- The Sensory evaluation section should be related and compared to the Electronic tongue analysis for a better discussion.

- Conclusions: Authors should avoid at all costs repeating results contained in R&D. Practical recommendations and specific substitutions could be given here, in addition to discussing the extension of the results to other possible uses. A better formulation or formulations should therefore be chosen, indicating specific uses. In addition, the feasibility of commercialization, from the point of view of local regularization, of the products analyzed in the study should be clarified.

Round 2

Reviewer 2 Report

The authors have provided answers to some of the requirements; however, the following remain unresolved:

- Only significant figures should be reported in Tables 1 and 2.

- All abbreviations in Table 1 remain unexplained, e.g., DIMP.

- The reasons for the choice of formulations reported in Table 1 remain unexplained.

- The authors do not clarify where the changes in their Response 5 are reflected.

- There is no evidence for Response 6 in the corrected manuscript.
